# Nondestructive Detection of Magnetic Contaminant in Aluminum Casting Using Thin Film Magnetic Sensor

**DOI:** 10.3390/s21124063

**Published:** 2021-06-12

**Authors:** Tomoo Nakai

**Affiliations:** Industrial Technology Institute, Miyagi Prefectural Government, Sendai 981-3206, Japan; nakai-to693@pref.miyagi.lg.jp; Tel.: +81-22-377-8700

**Keywords:** magnetic thin film, magneto-impedance sensor, high frequency, surface normal magnetic field, nondestructive inspection, tool steel chipping

## Abstract

The thin film magneto-impedance sensor is useful for detecting a magnetic material nondestructively. The sensor made by single layer uniaxial amorphous thin film has a tolerance against surface normal magnetic field because of its demagnetizing force in the thickness direction. Our previous study proposed the sensitive driving circuit using 400 MHz high frequency current running through the sensor to detect the logarithmic amplifier. We also confirmed the sensitivity of the sensor within 0.3 T static normal magnetic field, which resulted in detection of 5 × 10^−8^ T of 5 Hz signal. This paper proposes a nondestructive inspection system for how detecting a contaminant of small tool steel chipping in aluminum casting specimen would be carried out. Three channel array sensors installed in the 30 mT static field detecting area were fabricated and experimentally showed a detection of low remanence magnetic contaminant in a bulk aluminum casing specimen.

## 1. Introduction

An inspection of all items in the manufacturing process is desirable for realizing the smart manufacturing system, which utilizes the sensor technology for manufacturing process [1,2]. A nondestructive inspection, which detects a contaminant of small tool steel chipping in a product, is important and it has been applied for various products including foods, medicines, and industrial resins [3,4,5]. In order to realize a high-sensitive and high-resolution inspection, special sensitive equipment is required, such as SQUID (Superconducting Quantum Interference Device) or atomic magnetometer [4] for magnetic sensors, and a microfocus X-ray tube or a CT (Computerized Tomography) system for X-ray imaging [3]. For application to the actual production line of a manufacturing factory, the following are required: the first is a reduction of cost of the inspection apparatus, the second is a space saving for installation based on the demand to apply for the existing production line, and the third is a tolerability against environmental changes around the installed production line. This restriction must be taken into consideration for proposing a new method of sensing to apply to an actual manufacturing system in order to realize the smart manufacturing system in the actual factory. A well-known method of such a detection is the search coil method, which detects a change of impedance of the exited coil caused by an existence of conductive or magnetic material. Although this search coil method is sensitive and has a simple structure, it is difficult to apply for detecting a magnetic chipping in the vicinity of conductive materials [6,7,8]. This is because an occurrence of eddy current in the conductive material causes a noise signal that overwhelms the sensing signal from the chipping; thus, the sensing signal would be hidden behind the eddy current signal. This situation occurs in cases such as detecting tool steel chipping in an aluminum casting, or the chipping on a printed circuit board having a copper conductive circuit pattern. This paper proposes a detection system for inspecting such specimens nondestructively, which is applicable for the actual conveyer line of manufacturing for realizing the inspection of all items.

A thin film magneto-impedance (MI) sensor [9,10,11,12,13,14] has high sensitivity, and it is useful for detecting magnetic small chippings nondestructively [15,16,17]. The sensor, which is made up of a single layer uniaxial amorphous thin film, has tolerance against surface normal magnetic field because of its demagnetizing force in the thickness direction. Our previous study proposed the sensitive driving circuit using 400 MHz high frequency current running through the sensor, which utilizes a logarithmic amplifier in a differential layout as a detector [18]. We confirmed the sensitivity of the sensor within 0.3 T static normal magnetic field, which resulted in realizing a detection of 5 × 10^−8^ T of 5 Hz signal [19]. A magnetic domain variation of the sensor element within the 0.3 T normal field was also observed as a function of measurement field along the in-plane direction. This study demonstrated that the single layer amorphous thin film within the 0.3 T surface normal field has a domain variation that is almost the same variation as when placed in a space without the strong normal field [20].

This paper proposes a method that is applicable for detecting a magnetic contaminant such as chipping in a conductive material. In order to prevent the appearance of eddy current noise from the conductive material, such as a bulk aluminum body or a conductive copper sheet, a static magnetic field is utilized instead of inductive alternating field. In our proposed detection system, the magnetic field sensor is placed inside the magnetic field for the purpose of detection of the small chipping at the same time of magnetization caused by the strong static magnetic field applied in the measurement area. The merits of this sensor layout are both detecting the magnetic chipping at the nearest position and also detecting the maximum magnetization at the same time as the chipping. These merits are realized in particular by using a thin film magneto-impedance sensor. Our sensor consists of a single layer amorphous soft magnetic strip that has a uniaxial magnetic anisotropy in the in-plane direction. The uniaxial easy axis is placed along the width direction, which is the short side direction of the sensor strip, and it is perpendicular to the current flowing along the length of the element. In this element, the magnetic moment is forced by both the uniaxial anisotropy and the shape anisotropy of the rectangular element. The magnetic domain structure in the element is determined to be an equilibrium state by an interaction between the applied external field and the anisotropy field of the element. The element impedance is strongly affected by the structure of magnetic moment, due to the dominant phenomenon of high frequency impedance for the magnetization rotation [2,21]. The structure of the moment is determined by the magnetic domain structure of the element at a certain external field. This operation principle of this soft magnetic single layer element leads to an important property of our element, which is an undistorted sensing ability, even after an extraordinary strong surface normal field is applied to the element, due to the very slight magnetic hysteresis. This property makes the element the first candidate of our proposed nondestructive inspection system.

In this study, a prototype sensing system with three sensors in a line array configuration was fabricated inside the 30 mT static magnetic field. Figure 1 shows a conceptual image of the measurement system. The system would be designed to be installed in the conveyor of a manufacturing system. Magnetic field would better be applied by magnets for preventing energy consumption, and thin film sensors are designed to place just above the magnet for keeping the sensing volume as large as possible. The following sections of this paper report the details of the development of this measurement system. The system consisted of both components of a thin film MI sensor array and a field generating unit of strong surface normal magnetic field. An experimental confirmation of the detection of small tool steel chipping inside an aluminum casting specimen is also reported.

## 2. Sensor and Systemization of Measurement System

The structure of the sensor element and the driving circuit were the same as the previous report [7,8,9]. In this paper, details of the sensor and a systemization trial are explained.

### 2.1. Sensor Element

The sensor element was fabricated by a thin-film process. An amorphous Co_85_Nb_12_Zr_3_ film was RF-sputter deposited onto a soda glass substrate and then micro-fabricated into rectangular elements by the lift-off process. The element was a single layer film and had a 1000 μm length, 50 μm width, and 2.1 μm thickness. The tens of elements are aligned in a parallel configuration and connected by Cu thin-film strips to form a meander pattern. A magnetic field was applied with the RF-sputter deposition for the purpose of inducing uniaxial magnetic anisotropy. The direction of the magnetic anisotropy in this study was along the width, so the short side direction of the element strip. It is induced along the direction of the magnetic field while sputtering. The fabrication process flow diagram is shown in Figure 2.

Figure 3 shows fabricated sensors on a glass substrate. The 7 × 7 sensors were made on a 26 × 26 × T1 mm soda glass substrate. These sensors in the photo consist of the differential configuration, where two sensors were adjacent with each other to configure a pair of differential ones. The system requirement of this research was a detection of contaminant of a small magnetic piece in an industrial product without overlooking. The differential sensor has a possibility to overlook a signal from the contaminant right above the sensor, as shown in Figure 4. In this case, if the signal obtained by each sensor element has the same magnitude and the same phase, then it is recognized as a common mode noise and rejected by the effect of differential signal processing. Due to this reason, only the right one of the sensor pair was used as a single sensor, not as a differential sensor, in this study.

Figure 5 shows a measurement system that evaluated the dependence of sensor impedance as a function of magnetic field applied in the sensing direction. The sensitivity of the thin film MI sensor element exists in the in-plane longitudinal direction on the element. The measurement system can evaluate the impedance of a single sensor element using a high frequency probe. The high frequency impedance was measured by network analyzer E5071C (Keysight Technologies, Santa Rosa, CA, USA) using reflection signal measurement (S11). The external magnetic field was generated by a Helmholtz coil, and the measurement was controlled by a laptop computer (PC).

Figure 6 shows a typical impedance variation of the sensor. Figure 7 shows a variation of sensitivity dZ/dH as a function of the external field. This is one of the elements that was used in our fabricated system. The measurement was carried out in various frequencies ranging from 300 MHz to 700 MHz. The measurement incident power of the high frequency signal was set at −10 dBm, which was the same as the actual incident power of fabricated system. Figure 6a shows a whole value of impedance *Z* = *R* + *Xi*, where *R* is resistance and *X* is reactance. The variation range of impedance increases as a function of frequency. The sensitivity dZ/dH has a maximum value of around 500 to 600 MHz due to a peak position shift outward that appears as increasing the frequency. Figure 6b shows a variation of resistance, and Figure 6c shows a variation of reactance of the same element. The variation of reactance shows an appearance of additional dipping points of variation profile near the sensor bias point as increasing the frequency. The sensor bias point was set at around the external field of 1.4 kA/m (17.5 Oe) because it was as a maximum sensitivity point. The maximum sensitivity point means that it is the maximum point of dZ/dH of the impedance profile. The appearance of this deformation of reactance is observed over 500 MHz; thus, our driving circuit that was designed for 400 MHz utilized a linier range of both the resistance and the reactance. The detection of the logarithmic amplifier of our proposed circuit is based on a differential detection of the actual waveform of 400 MHz signal as a differential input, where the variation of both the amplitude and the phase is sensitive for the output. The setting of the linear variation region of both resistance and reactance was supposed to be desirable for our driving circuit. The details of dependence of the sensing linearity and the driving frequency would be the subject of another report.

Figure 8a–c show impedance variation profiles of each sensor used in the prototype system fabricated in this study. Three sensors were installed with the distance of 60 mm between the sensors. These sensors were fabricated on a same glass substrate, as shown in Figure 3, and after dicing was processed, these sensors were evaluated and used for the systemization. The environmental tolerance of the sensor is shown in Appendix A.

### 2.2. Sensor Combined with Driving Circuit

Figure 9 shows a view of the sensor head unit. The sensor element was fixed on the tip of a printed circuit board (PCB), and the PCB was fixed on an aluminum plate designed to be suitable for mounting our apparatus. The transmission line of PCB was designed as it has a high frequency impedance of 50 Ω, and SMB connectors were connected on the end. The substrate of PCB was a glass epoxy substrate (FR4). The 400 MHz current was flowing through the sensor coming from one connector and then going out to the other connector. This sensing head was electrically connected to the unit of the driving circuit by SMA cable.

Figure 10 shows a view of an evaluation unit of the sensor unit performance. The sensor unit consisted of a sensor head, Figure 9, and a driving circuit unit. The output signal from the sensor unit was measured by a digital oscilloscope. The digital oscilloscope used in this research had 12-bit resolution for digitizing. The evaluation of the sensor unit was carried out by varying the external magnetic field simultaneously with the observation of the output voltage using the oscilloscope. The sweep waveform of the external magnetic field was a triangular waveform with the frequency of 0.02 Hz, which was generated by a function generator and a power amplifier. The range of external magnetic field was from −30 Oe (−2.4 kA/m) to +30 Oe (+2.4 kA/m), and it led to an output voltage in the range of almost 7 V. The details of this driving circuit were reported in reference [7].

Figure 11 shows variations of the output of the sensor unit as a function of the applied external field. These measurements correspond to the sensors measured the impedance in Figure 8. Each sensor unit has almost the same properties with a very slight hysteresis at the bias point of measurement.

### 2.3. Magnetic Field Generating Unit with Installation of Sensors

Figure 12 shows a schematic explanation of the measurement system. A square-shaped iron magnetic core forms a base unit, and four NdFeB magnets were fixed on each side of the square-core. Each magnet had a flat parallelepiped rectangular shape, and magnetic poles existed on the flat face of the rectangular magnet. The upper and bottom magnets in this figure were placed at the opposite poles and were facing each other. This magnetic structure of Figure 12 generates a magnetic field in the vertical direction. The magnetic field around the edge of the rectangular magnet has a disorder structure, in particular, around the gap of the adjacent two magnets. The measurement system requires the sensor head to be placed on the magnet surface for the purpose of keeping the measurement area wide; therefore, the existence of the disorder magnetic field was undesirable. In order to settle this problem, homogenization plates for magnetic flux were installed. In this study, 200 × 120 × T12 mm iron plates made by S45C were placed on each side of the magnetic structure.

The thin film MI sensor needs a bias magnetic field for operation, due to the maximum dZ/dH point of the impedance profile that existed at a certain magnetic field value, as shown in Figure 8. The bias point of sensors in this study was approximately 1.4 kA/m (17.5 Oe). In this magnetic apparatus, the bias field was obtained by a magnetic field fluctuation on the magnetic homogenization plate. The magnetic field has a tendency of expanding as the distance from a magnet increases, as shown in the side view of Figure 12. By utilizing this effect, the bias point of each sensor was made to be controllable along the tangential position Y on the homogenization plate.

Figure 13 shows a photo of the fabricated measurement system. The sensors are placed on the upper surface of the bottom homogenization plate with 60 mm distance between sensors. The sensing direction of the sensors were placed in the tangential direction of the homogenization plate. This means that the vertical magnetic field generated by the magnets was in the surface normal direction of the thin film sensor.

Figure 14 shows a measured distribution of the magnetic field in the vertical direction of the system. The measurement was carried out using a Hall-probe Tesla-meter, Model 5080 (F.W. BELL, Milwaukie, OR, USA). The probe was scanned 2-dimensionally on Y-Z plane, keeping the measurement direction in Z. Figure 14a shows the color map of magnetic flax density at the central vertical plane of the unit, which was parallel to the Y-Z plane. The horizontal axis indicates the position along the sample feeding direction. Figure 14b shows the contour diagram. The dashed line indicates a horizontal middle line, which is a vertically central line of the square-core. Based on this result, the magnetic flux density of the measurement area ranged from 24 mT to 33 mT in the vertical direction. Sensors were installed just above the bottom homogenization plate, and the vertical field here was 33 mT. The thin film MI sensor in this study had a sensitivity of more than 5 × 10^−8^ T for the 5 Hz signal [8]. This sensitivity is enough for this system to be able to measure an actual waveform of the magnetic field when a magnetic small chipping is running through the measurement area.

Figure 15 shows the whole view of the developed prototype system. The three sensors were driven by stacked three driving units, and the output signal was measured and memorized by the multi-channel oscilloscope. The trigger of measurement was done by a photo-sensor, which recognizes the incident timing by detecting a certain position of the feeding sample. Our driving circuit has a simple structure and consisted of only 6 units. The size of the driving unit is expected to be designed smaller in the future by developing an integrated PCB board. Although this system uses a high frequency driving signal and a highly sensitive detection circuit that utilizes the actual raw waveform inputted to a differential logarithmic amplifier, it would be expected to develop a smaller and more sophisticated sensor system.

## 3. Results and Discussion

In this section, a trial of actual measurement of a contaminant of tool steel chipping in an aluminum casting specimen is indicated.

Figure 16a shows a schematic of the measured sample. Figure 16b shows a photo of the sample. An aluminum casting body with a complex shape was prepared and a *φ*5 × H5 mm tool steel column was installed near the center of the body using the central drill-hole. The sample simulated an aluminum component with a contaminant, such as a broken drill chipping. Figure 17 shows a photo of the measurement. The aluminum specimen was placed obliquely, and moved from the back to this side of the measurement system. The sensor head was covered by non-magnetic conductive plate for preventing electrostatic noise.

Figure 18 shows a measured magnetization curve (MH-loop) of the tool steel column. This MH-loop shows that this chipping has a soft magnetic property with low remanence. Our proposed measurement method is advantageous for detecting such low remanence species, due to the measurement principle of detecting the chips simultaneously with magnetization using a strong magnetic field.

The magnetic field generated by a magnetized magnetic small piece would be estimated using the equation of magnetic dipole, as in Equation (1).
(1)$$H=-\frac{1}{4{\pi \mu }_{0}}\nabla \frac{m\cdot r}{r^{3}}$$
where ***m*** is a vector of magnetic moment placed at the origin of coordinate-system, ***r*** is a position vector for estimating the magnetic field, and *μ*_0_ is a vacuum permeability.

A magnetic flux density in the Y direction at the position ***r*** is estimated as shown in Equation (2).
(2)By=μ0Hy=−3myz4π(x2+y2+z2)52

In this study, the Y direction corresponded to the sensing direction of the thin film MI sensor. The measurement specimen was fed in a constant velocity along the parallel line of the Y axis. Then, a variation of sensor signal obtained by our sensor system was estimated by Equation (2).

Figure 19 shows a feeding position of the contaminant of tool steel chipping in the aluminum casting body as shown in Figure 16. The measurement was carried out with the feeding velocity of 70 mm/s.

Figure 20 shows an estimated profile of the sensing signal of three sensors. The distances from the scanning line correspond to the distance between the sensor and the scanning line of the chipping for the three sensors, respectively, as shown in Figure 18. It was estimated in a condition when the sensor was placed at Y = 0.

Figure 21 shows a measured result using the prototype system and actual aluminum casting specimen. The feeding position of the tool steel chipping also corresponded to Figure 19. The sensitivity of this sensor unit was at 0.1 V output when 0.1 μT (1 mG) was applied. The peak value of 1.6 V for CH3 sensor output correspond to 1.6 μT (16 mG). In comparison between the estimation, Figure 20, and the measurement, Figure 21, the tendency of the profile corresponds approximately. The actual sensors had variation of characteristic. In this study, sensitivity correction was not carried out, and the difference between the measurement and the estimation is supposed to come from the sensitivity variation of each sensor. The sensor position also varied due to the adjustment of the bias point of the sensors. The bias point adjustment was controlled by the sensor position along the Y direction. It utilizes a magnetic field fluctuation as a function of Y position just above the homogenization plate. Each sensor has its own bias point, and the actual magnetic generating structure has a certain assembling error. The difference of zero-cross point of the measurement result in Figure 21 is supposed to come from these reasons. A trial of estimating 3-dimensional feeding position based on the sensing signal of this measurement is now being researched. The accuracy of the feeding position of the chipping should be improved as this investigation is developing.

Figure 22 shows a dependence of output range as a function of feeding in the X-position. The feeding height was set at a constant value of 40 mm, which was the same as Figure 21. The variation range in this figure means a range between the maximum and minimum of the output signal of the measurement signal profile. This result shows that the 60 mm distance sensors installed in the field generator can detect the chipping without overlooking.

## 4. Summary

The proposal of a nondestructive inspection system for detecting a magnetic contaminant such as a small tool steel chipping with low remanence in a conductive aluminum casting specimen was carried out. The developed system utilized thin film magneto-impedance sensors installed in a strong magnetic field in the surface normal direction of the sensor. The prototype system was fabricated and consisted of three channel thin film sensors having array configuration installed in a detecting area that had a 30 mT vertical static field. The tolerance of the thin film magneto-impedance sensor against the strong normal field has been confirmed previously. This article reports a systemized trial of a nondestructive inspection for detecting a contaminant of tool steel chipping in conductive aluminum casting specimen by utilizing this sensor characteristic. The vertical magnetic field generated by NdFeB magnets was controlled using a magnetic field homogenization plate to form a parallel field along the surface normal direction of the installed thin film sensor. As a result, a detection of a low remanence magnetic contaminant of tool steel chipping in a bulk aluminum casting specimen was experimentally confirmed, and the wave form of the feeding specimen was clearly observed by the neighboring sensors in a distance of 60 mm without overlooking. The systemization of an inspection system using a thin film magneto-impedance sensor having a strong tolerance against the surface normal field was carried out, and the effectiveness of the inspection system was confirmed.

## 5. Patents

Japanese patent 6842164 have been obtained based on this work.

## Figures and Tables

**Figure 1 sensors-21-04063-f001:**
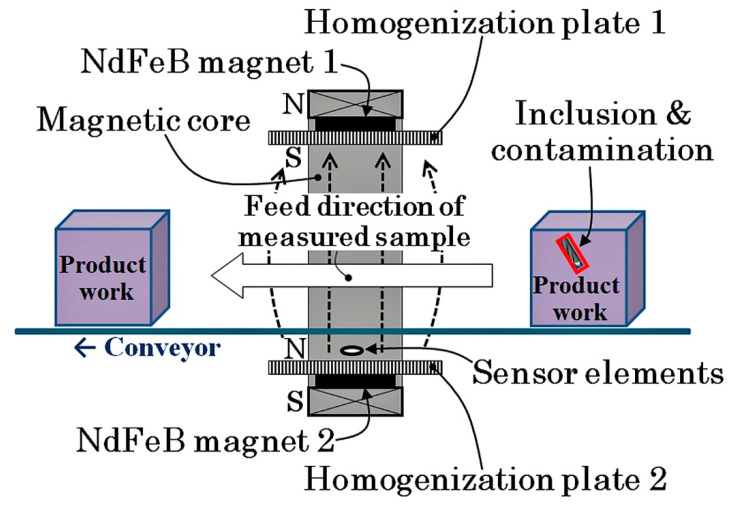
Conceptual image of development.

**Figure 2 sensors-21-04063-f002:**
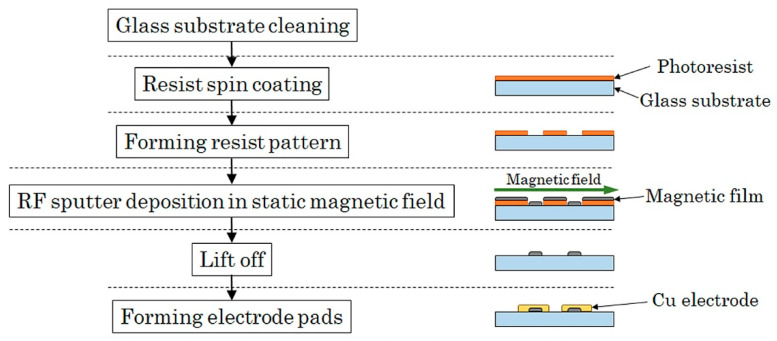
Fabrication process flow of sensor.

**Figure 3 sensors-21-04063-f003:**
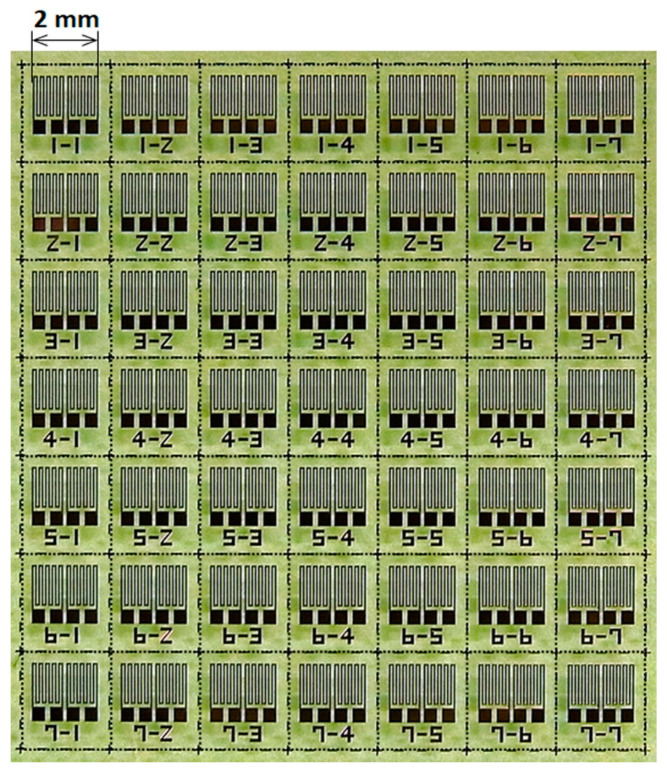
Fabricated sensors on glass substrate.

**Figure 4 sensors-21-04063-f004:**
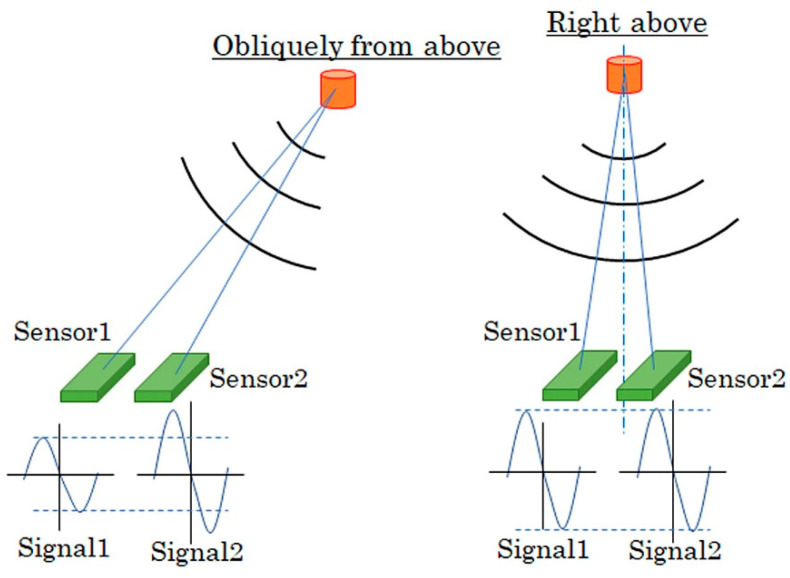
Schematic of a demerit of differential sensor.

**Figure 5 sensors-21-04063-f005:**
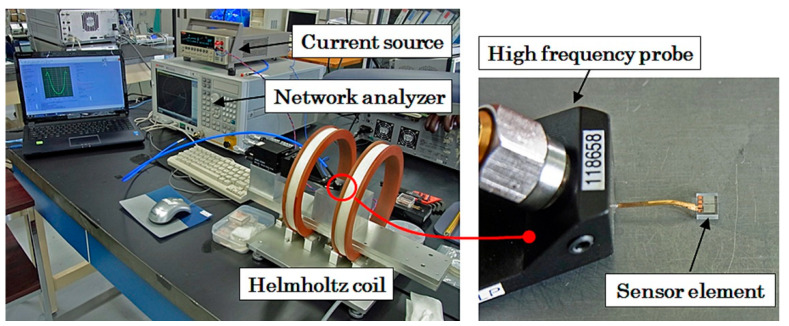
Measurement system of sensor element impedance.

**Figure 6 sensors-21-04063-f006:**
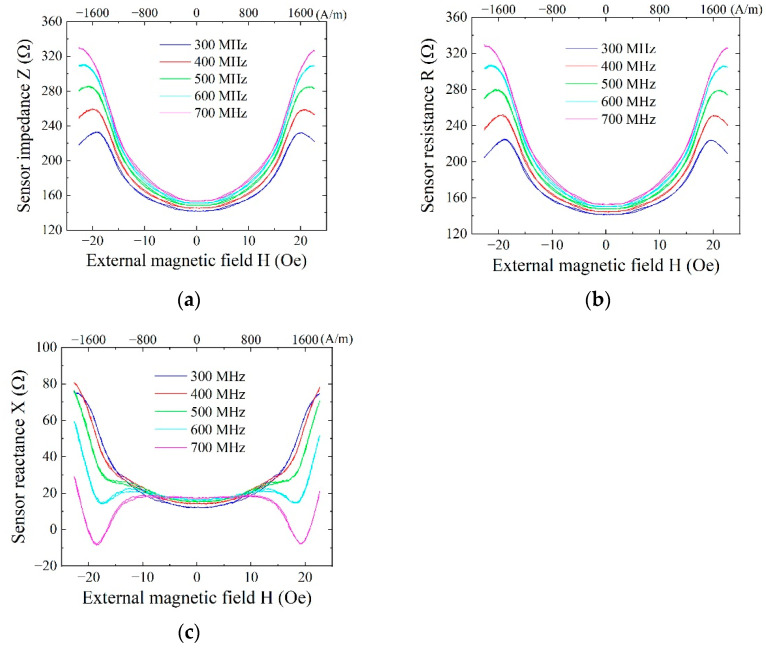
Typical impedance variation of the sensor based on *Z* = *R* + *Xi*; (**a**) Impedance *Z*; (**b**) Resistance *R*; (**c**) Reactance *X*.

**Figure 7 sensors-21-04063-f007:**
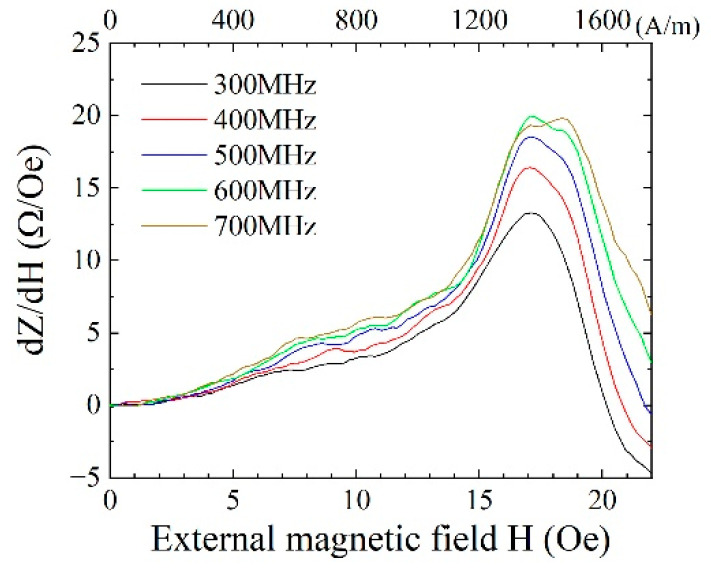
Typical variation of sensitivity dZ/dH of the sensor.

**Figure 8 sensors-21-04063-f008:**
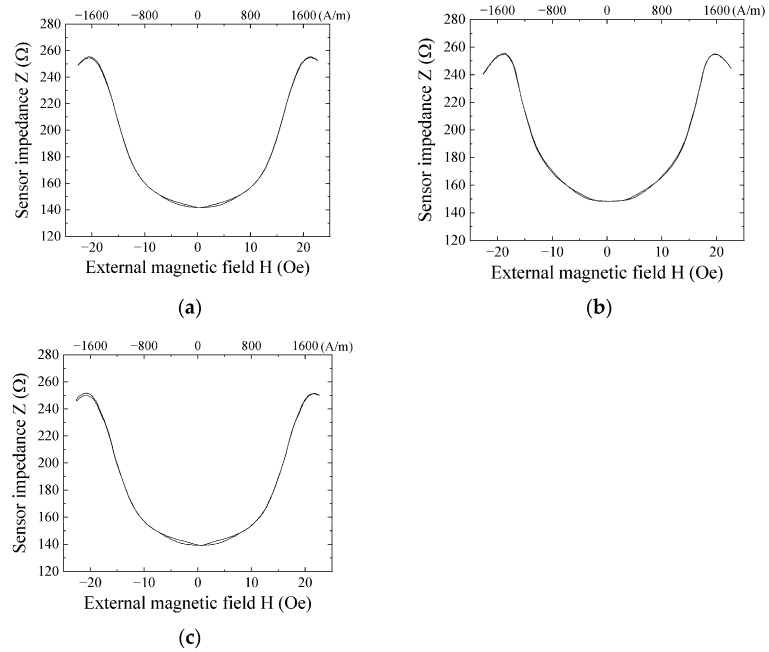
Impedance variation of sensors used in the prototype system; (**a**) Sensor 1; (**b**) Sensor 2; (**c**) Sensor 3.

**Figure 9 sensors-21-04063-f009:**
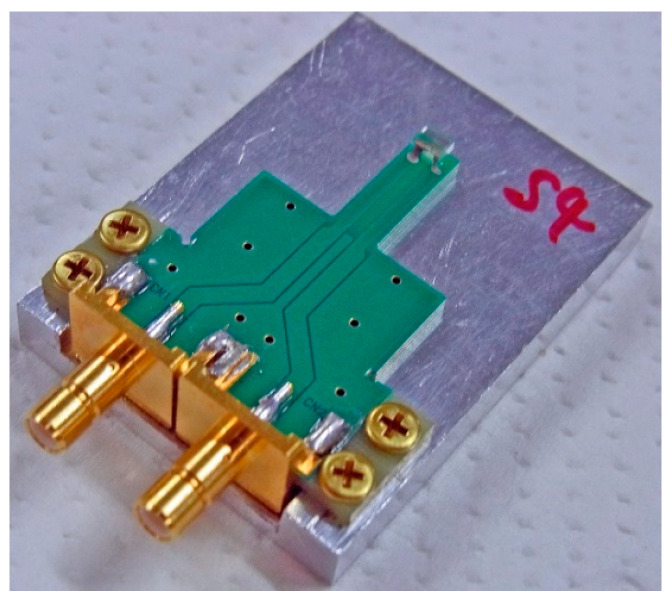
View of sensor head unit.

**Figure 10 sensors-21-04063-f010:**
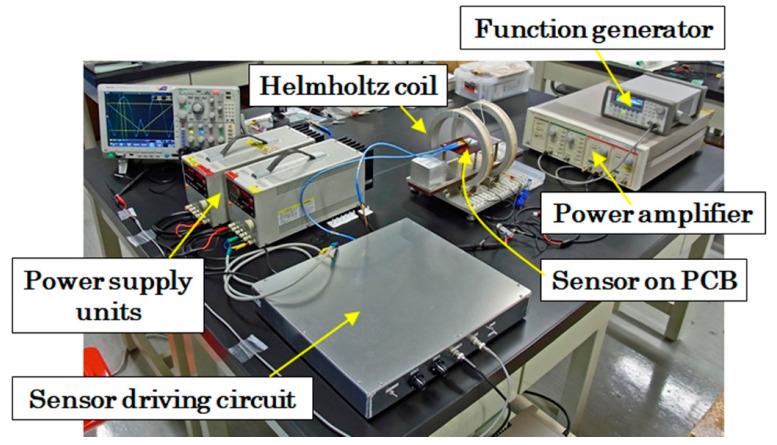
View of evaluation unit of the sensor unit performance.

**Figure 11 sensors-21-04063-f011:**
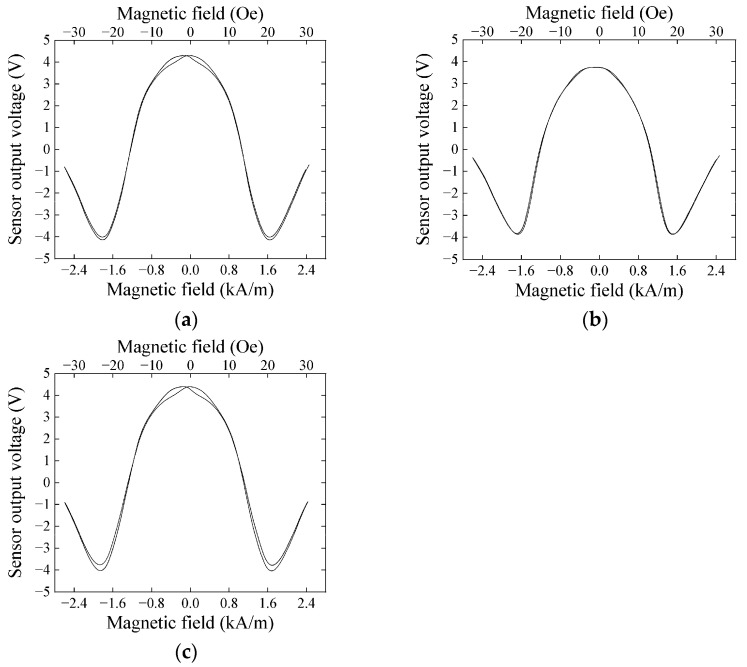
Variations of the output of the sensor unit; (**a**) Sensor 1; (**b**) Sensor 2; (**c**) Sensor 3.

**Figure 12 sensors-21-04063-f012:**
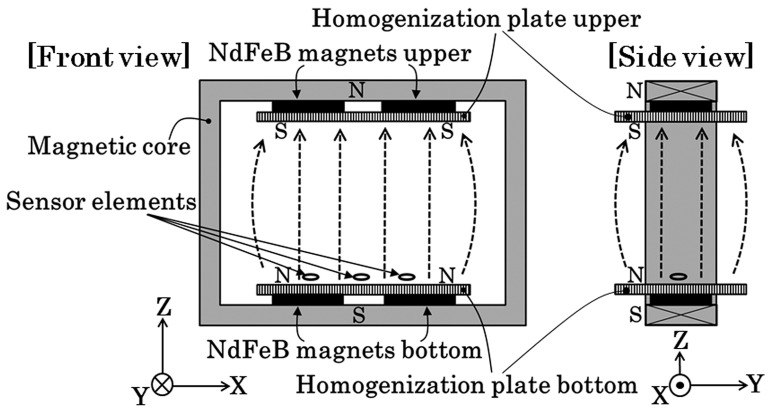
Schematic of measurement system.

**Figure 13 sensors-21-04063-f013:**
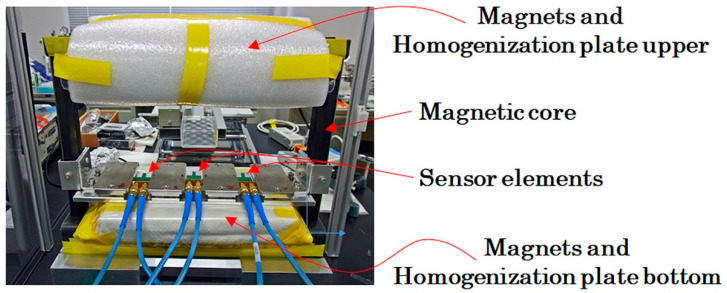
Photo of fabricated measurement system.

**Figure 14 sensors-21-04063-f014:**
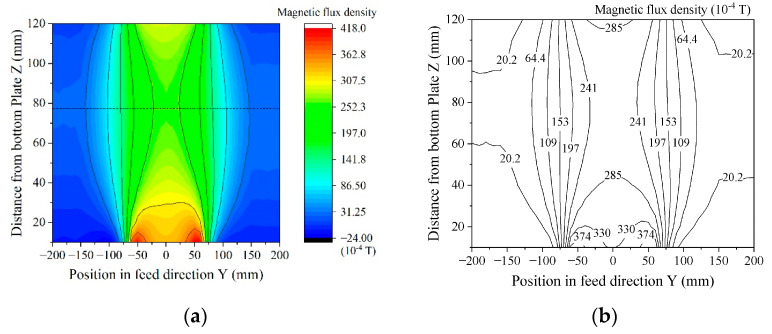
Measured distribution of magnetic flux density of the system; (**a**) Color map; (**b**) Contour diagram.

**Figure 15 sensors-21-04063-f015:**
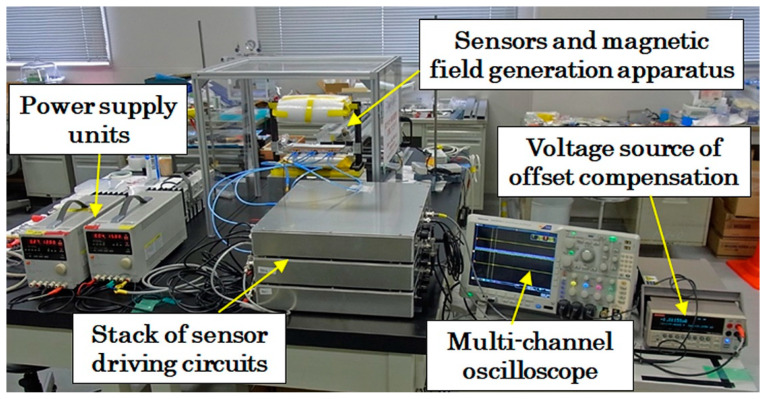
Whole view of the developed prototype system.

**Figure 16 sensors-21-04063-f016:**
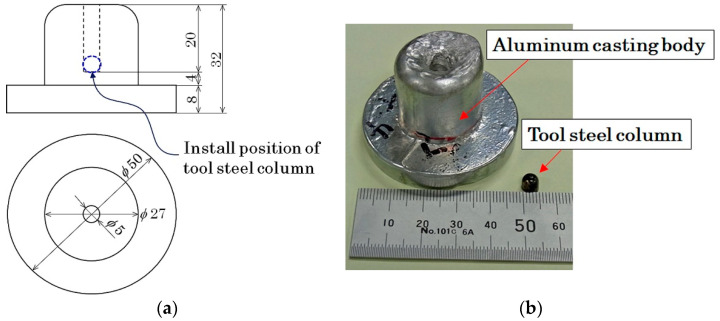
Schematic of measured sample; (**a**) Dimensions drawing; (**b**) Photograph.

**Figure 17 sensors-21-04063-f017:**
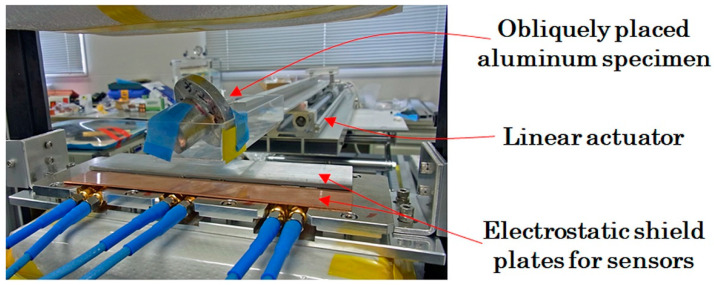
Photo of measurement while feeding.

**Figure 18 sensors-21-04063-f018:**
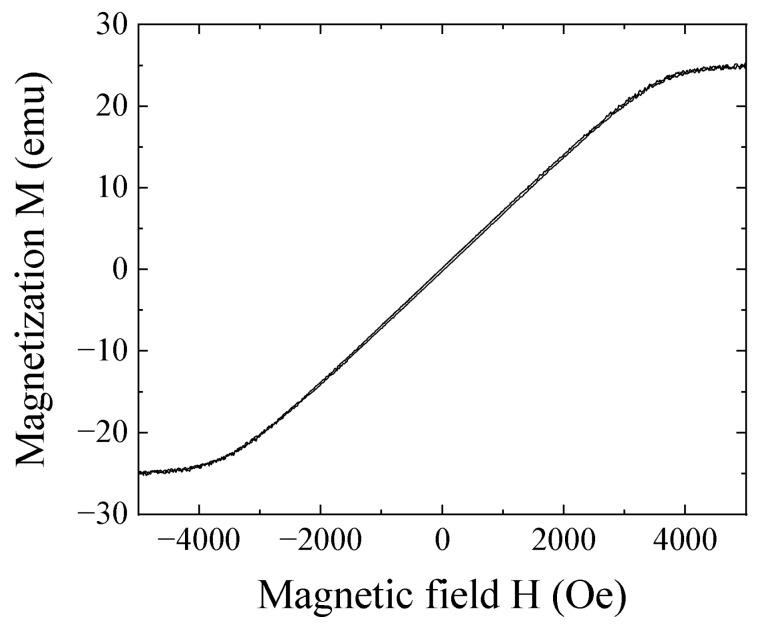
Magnetization-loop of the tool steel chipping.

**Figure 19 sensors-21-04063-f019:**
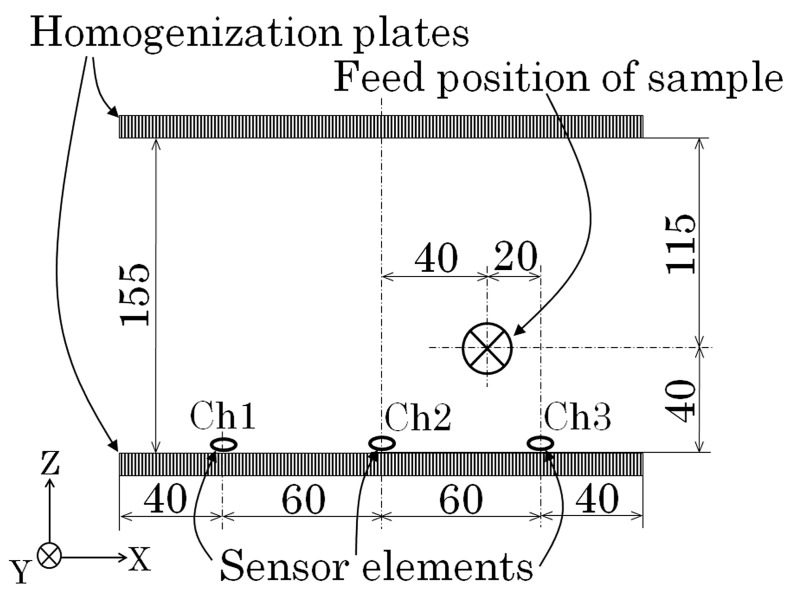
Feeding position of the tool steel chipping.

**Figure 20 sensors-21-04063-f020:**
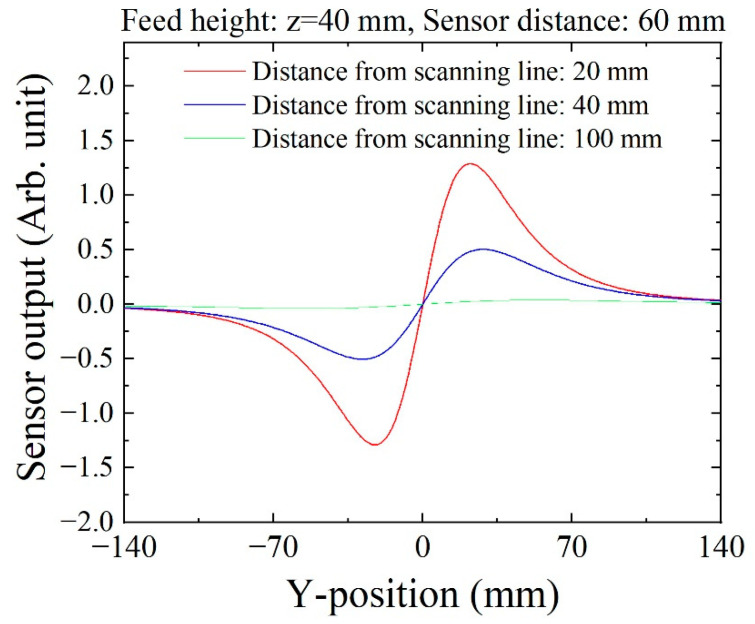
Estimated profile of sensor signal for three sensors in different positions.

**Figure 21 sensors-21-04063-f021:**
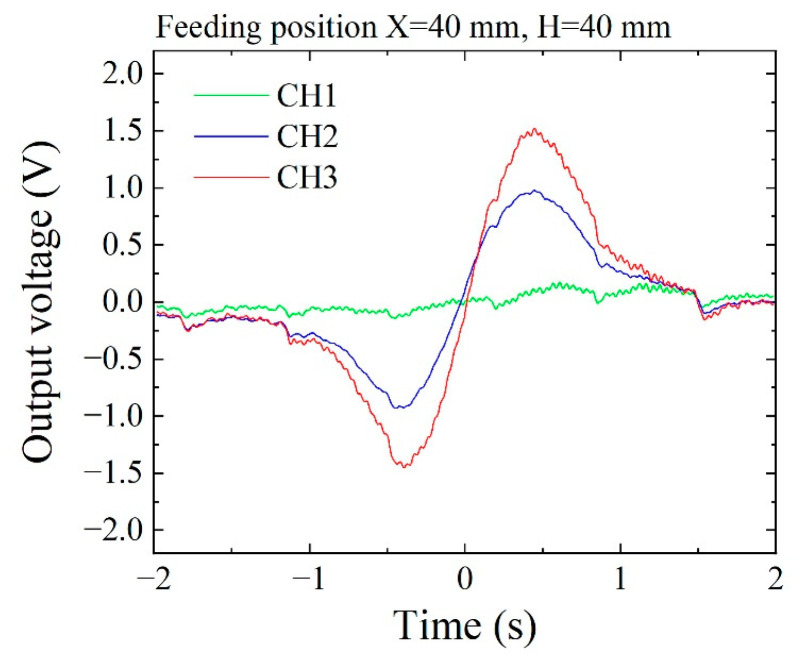
Measured profile of sensor signal obtained from the prototype system.

**Figure 22 sensors-21-04063-f022:**
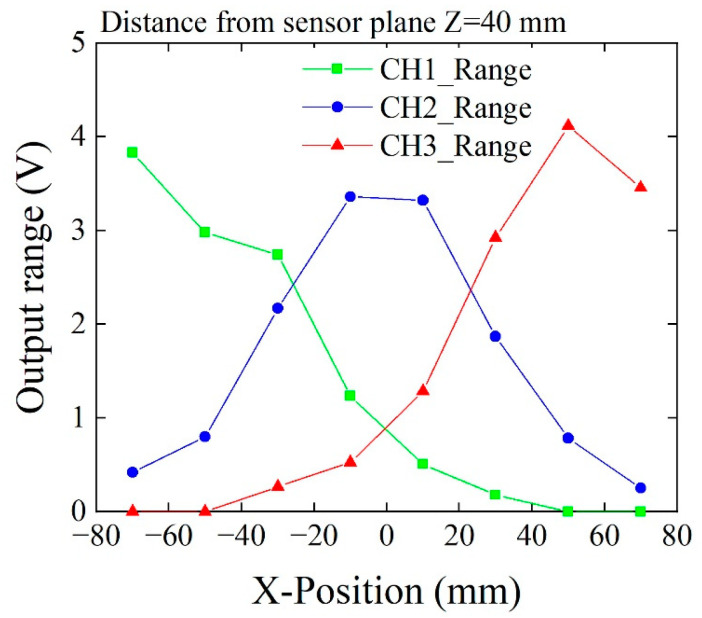
Variation of output range as a function of feeding position.

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
