# Peer review of "Nondestructive Detection of Magnetic Contaminant in Aluminum Casting Using Thin Film Magnetic Sensor"

_sensors, 2021, doi:10.3390/s21124063_

Round 1

Reviewer 1 Report

This paper indicates a thin film magneto-impedance sensor to nondestructively detect magnetic material in aluminum test piece. The paper is mainly focused on the detail investigation of characteristic of the developed sensor, which has single layer uniaxial amorphous thin film. The obtained results are interesting and looks reasonable.  My comments are as follows.

  • Fig.11, 14 and 15 need labels indicating items as same as Fig.13.
  • Please indicates the install position of tool steel column in Fig. 14 so that the reader easily understand the contents of the paper.

Reviewer 2 Report

This is a very comprehensive and well presented body of work. I commend the author for this study. My only suggestion is to improve the English language and quality of explanation in some places.

  • I found the use of the word 'included' in the title confusing. When something is 'included' it is usually deliberate or part of a design. What is meant here is that the steel chipping is a contaminant or foreign matter. I suggest a change to something like 'Nondestructive detection of magnetic contaminant in Aluminum casting using a thin film magnetic sensor'.
  • elsewhere in the text e.g. line 49, this use of 'included' may cause the same confusion. 
  • line 63 - what is 'magnetic momentum'? - do you mean magnetic moment?
  • line 71 - you claim the sensing ability is undistorted even with a strong perpendicular field. Can you explain what happens if the strong applied field is not exactly at 90 degrees e.g. 89 degrees - is this a problem? How much misalignment can be tolerated?
  • line 114 - 'reduces the sensing signal coming from long distance' - this seems a strange statement, I understand a sensor can reject a signal at a certain direction of field, but how can it differentiate a source distance?
  • line 149 - sensor bias point set at 20 Oe. Line 225 bias point is 17.5 Oe. Is this a contradiction?
  • Is it useful to have a plot of dZ/dH?

Reviewer 3 Report

The article is devoted to the study of the relevant problem of detecting adverse inclusions from tool steel in aluminum products, which may appear in the technological process during microscopic destruction of a steel machining tool. In the article under review, such a detection is proposed to be carried out by the magnetic leakage field of a ferromagnetic inclusion that appears around inclusion as a result of magnetization by an artificial constant field with an induction of the order of 30 mT.

         The authors have proposed, manufactured and successfully tested a prototype of a detecting device (flaw detector), which includes a system of permanent magnets with a magnetic core and three thin-film magnetic field sensors of their own production with appropriate electronics.  The article describes the main stages of manufacturing thin-film sensors, as well as the results of testing their conversion characteristics. The process of testing a prototype flaw detector for detecting artificial defects is described in sufficient detail and convincingly. The design features of the described device are protected by a Japanese patent. The article is recommended for publication.

Author Response

Thank you for your recommendation for publication.

Reviewer 4 Report

The author presents nondestructive inspection of tool steel chipping included in 2 Aluminum casting using thin film magnetic sensor. I recommend the paper for publication in sensors Journal after the following comments are incorporated in the revised version of the paper.

1-The literature review is too short and not detailed. I think that an accurate discussion of the literature could substantially improve the introduction.

2- Please use W for resistance unit instead of Ohm.

3- Please explain in details the novel contribution of the paper in relation to previous published work (example ref. 7)

4- The author state that results shown in Fig 18 and Fig 19 are in good agreements. However, a big difference can be observed for CH2. Please give an explaination
